# Heterogeneous Decision Making in Mixed Traffic: Uncertainty-aware Planning and Bounded Rationality

Hang Wang[1], Qiaoyi Fang[1], Junshan Zhang[1]
[1]University of California, Davis
{whang,qyfang,jazh}@ucdavis.edu

The past few years have witnessed a rapid growth of the deployment of automated vehicles (AVs). Clearly, AVs and human-driven vehicles (HVs) will co-exist for many years, and AVs will have to operate around HVs, pedestrians, cyclists, and more, calling for fundamental breakthroughs in AI designed for mixed traffic to achieve mixed autonomy. Thus motivated, we study heterogeneous decision making by AVs and HVs in a mixed traffic environment, aiming to capture the interactions between human and machine decision-making and develop an AI foundation that enables vehicles to operate safely and efficiently. There are a number of challenges to achieve mixed autonomy, including 1) humans drivers make driving decisions with bounded rationality, and it remains open to develop accurate models for HVs' decision making; and 2) uncertainty-aware planning plays a critical role for AVs to take safety maneuvers in response to the human behavior. In this paper, we introduce a formulation of AV-HV interaction, where the HV makes decisions with bounded rationality and the AV employs uncertainty-aware planning based on the prediction on HV's future actions. We conduct a comprehensive analysis on AV and HV's learning regret to answer the questions: 1) *How does the learning performance depend on HV's bounded rationality and AV's planning*; 2) *How do different decision making strategies impact the overall learning performance*? Our findings reveal some intriguing phenomena, such as Goodhart's Law in AV's learning performance and compounding effects in HV's decision making process. By examining the dynamics of the regrets, we gain insights into the interplay between human and machine decision making.

## 1. Introduction

Automated vehicle (AV) is emerging as the fifth screen in our everyday life, after movies, televisions, personal computers, and mobile phones [1, 2]. Their potential impact on safety and economic efficiency is substantial [3–6]. For instance, the National Highway Traffic Safety Administration (NHTSA) reported that preventable crashes in the United States caused $871 billion in economic and societal losses in 2010—approximately 1.9% of the GDP. While over 30 U.S. states have enacted AV legislation and AI-equipped vehicles continue to advance, experts acknowledge significant technical challenges remain [7–9]. Perhaps the most fundamental challenge is achieving both safety and efficiency in mixed-traffic environments, as AVs must coexist with human-driven vehicles (HVs), pedestrians, cyclists, and other road users for the foreseeable future.

The complicated interactions between HVs and AVs could have significant implications on the traffic efficiency given their different decision making characters. As such, a fundamental understanding on the heterogeneous decision making in the interplay, especially the impact of HVs' decision making with bounded rationality on AVs' performance, is crucial for achieving efficient mixed autonomy.

Existing works on modeling the interaction between AV and HV largely fall within the realm of conventional game formulation, in which both agents try to solve the dynamic game and adopt Nash equilibrium strategies [10–13]. This line of formulation faces the challenge of prohibitive computational complexity [14]. Needless to say, the decision making of HV and AV are different by nature. As supported by evidence from psychology laboratory experiments [15–17], human

Second Conference on Parsimony and Learning (CPAL 2025).

decision-making is often *short-sighted* and deviates from Nash equilibrium due to their *bounded rationality* in the daily life [18–20]. In particular, HV's bounded rationality is unknown a prior and it remains challenging to develop an accurate model for HV's decision making. As a result, it is sensible for AVs' decision making to leverage *uncertainty-aware planning* for safety maneuvers in response to human behavior [21, 22]. Clearly, the heterogeneous decision making by HVs and AVs exposes intrinsic complexities in the mixed autonomy.

Along the line of [13, 23], we consider a two-agent system with one AV and one HV, where the HV takes the action by planning for a short time horizon, and the decision-making is sub-optimal and noisy due to bounded rationality. The AV utilizes uncertainty-aware lookahead planning based on predictions of the HV's future actions. The primary objective of this study is to understand the performance of heterogeneous decision making in the mixed autonomy by answering the following questions: *1) How does the learning performance depend on HV's bounded rationality and AV's planning? 2) How do different decision making strategies between AV and HV impact the overall learning performance?*

The main contributions of this paper can be summarized as follows:

**(1) We first focus on the characterization of the regrets for both the HV and the AV, based on which we identify the impact of *bounded rationality* and *planning horizon* on the learning performance.** In particular, we present the upper bound on the regret, first for the linear system dynamics model case and then for the non-linear case. We start with the linear case, and show the accumulation effect due to the AV's prediction error and its impact on AV's learning performance. Building on the insight from the linear case, we model the prediction error as a diffusion process in the non-linear case to capture the accumulation effect. By studying the upper bound, we identify the compounding effects in HV's decision making due to bounded rationality and the Goodhart's law in AV's decision making associated with the planning horizon.

**(2) We study the impact of HV's bounded rationality on the overall learning performance and the regret dynamics of AV and HV.** We first establish the upper bound on the regret of the overall system due to HV's bounded rationality and AV's uncertainty-aware planning. Our regret bound naturally decompose into two parts, corresponding to the decision making of AV and HV, respectively. We examine the regret dynamics of the overall system theoretically and show how do different learning strategies between AV and HV affect the learning performance during each individual interaction through empirical study. The experiments details are available in Appendix F.

## 2. Related Work

**Mixed Autonomy.** Prior work on mixed autonomy traffic has primarily focused on specific dynamics models and empirical studies. For instance, [24] uses Bando's model for vehicle behavior analysis, while [25] studies AV's impact on HV driving volatility using predetermined AV acceleration models. The human factor has been examined through high-fidelity driving simulators [26], and stochastic models have been proposed to capture human behavior uncertainty [27]. On the learning side, [4] demonstrates congestion reduction using deep RL under the intelligent driver model (IDM). Without imposing specific models on HV and AV's decision making dynamics, our work focuses on the performance of different learning strategies in the mixed autonomy.

**HV-AV Interaction Model.** For modeling HV-AV interactions specifically, several game-theoretic approaches have been proposed. [10] and [13] use Stackelberg and two-player game formulations respectively, while [12] develops a hierarchical planning scheme. Although [23] attempts to address game formulation limitations using underactuated dynamical systems, it assumes identical decision-making horizons for both vehicle types. While related fields like ad-hoc team problems [28] and zero-shot coordination [29] provide empirical insights, they focus on either cooperative scenarios or self-play robustness. Our work differs by analyzing the interaction between agents with different decision-making strategies without assuming cooperation, particularly examining the impact of opponent modeling errors on learning performance [30]. Despite the rich empirical results in the related field, e.g., Ad-hoc team problem and zero-shot coordination, we remark that the theoretical

analysis on the interaction between AV and HV is still lacking, especially considering their different decision making. Moreover, our work deviates from the conventional game setting and aims to takes steps to quantify the impact of AV and HV's different decision making on the traffic system.

**Model-based RL.** MBRL with lookahead planning has shown promise in real-world applications due to its data efficiency [31]. Recent works [32, 33] utilize lookahead policies with future rollouts and terminal value functions to optimize action sequences. While [32] provides sub-optimality analysis under approximate models, our work differs fundamentally: we assume AV has access to environment dynamics but faces uncertainty from HV's bounded rationality. Moreover, our theoretical analysis uses regret with dynamically updating value functions, requiring significant different analytical techniques than previous work [32–34].

# 3. Preliminary

**Stochastic Game.** We consider the Stochastic Game (SG) defined by the tuple $\mathcal{M} := (\mathcal{X}, \mathcal{U}_A, \mathcal{U}_H, P, r_A, r_H, \gamma)$ [35], where $\mathcal{U}_A$ and $\mathcal{U}_H$ are the action space for AV and HV, respectively. Meanwhile, we assume the action space for HV and AV are with the same cardinality $M$ and let $\mathcal{U} = \mathcal{U}_A \times \mathcal{U}_H$. We denote $\mathcal{X}$ as the state space that contains both AV and HV's states. $P(x'|x, u_A, u_H) : \mathcal{X} \times \mathcal{U} \times \mathcal{X} \to [0,1]$ is the probability of the transition from state $x$ to state $x'$ when AV applies action $u_A$ and HV applies action $u_H$. $r_H(x, u_A, u_H) : \mathcal{X} \times \mathcal{U} \to [0, R_{\max}]$, $r_A(x, u_A, u_H) : \mathcal{X} \times \mathcal{U} \to [0, R_{\max}]$ is the corresponding reward for HV and AV. $\gamma \in (0,1)$ is the discount factor. We denote the AV's policy by $\pi : \mathcal{X} \times \mathcal{U}$ and use $\hat{u}_H(t)$ to represent AV's prediction on HV's real action $u_H(t)$ at time step $t$. We use $\rho_0$ to represent the initial state distribution.

**Value Function.** Given AV's policy $\pi$, we denote the value function $V^\pi(x) : \mathcal{X} \to \mathbb{R}$ as

$$\mathbf{E}\left[\sum_{t=0}^{\infty} \gamma^t r_A(x(t), u_A(t), u_H(t)) | x(0) = x, u_H(t)\right],$$

to measure the average accumulative reward staring from state $x$ by following policy $\pi$. The expectation is taken over $u_A(t) \sim \pi$ and $x(t+1) \sim P(x, u_H(t), u_A(t))$.

We assume the maximum value of the value function to be $V_{\max}$. We define $Q$-function $Q^\pi(x, u_A, u_H) : \mathcal{X} \times \mathcal{U} \to \mathbb{R}$ as $Q^\pi(x, u_A, u_H) = \mathbf{E}_\pi[\sum_{t=0}^{\infty} \gamma^t r_A(t) | x(0) = x, u_A(0) = u_A, u_H(0) = u_H]$ to represent the expected return when the action $u_A$, $u_H$ are chosen at the state $x$. The objective of AV is to find an optimal policy $\pi^*$ given HV's action $u_H$ such that the value function is maximized, i.e.,

$$\pi^* = \arg\max_\pi \mathbf{E}_{x \sim \rho_0, u_A \sim \pi(\cdot|x, u_H)}[Q^\pi(x, u_A, u_H)]. \tag{1}$$

Similarly, the objective function can be written as $\mathbf{E}_{x \sim \rho_0}[V^\pi(x)]$.

**Notations.** We use $\|\cdot\|$ or $\|\cdot\|_2$ to represent the Euclidean norm. $\|\cdot\|_F$ is used to denote Frobenius norm. $\mathcal{N}(\mu, \sigma^2)$ is the normal distribution with mean $\mu$ and variance $\sigma^2$. $I$ is an identity matrix.

## 3.1. Modeling AV-HV Interaction: Heterogeneous Decision Making

In this section, we examine in detail the interaction between one AV and one HV in a mixed traffic environment. More specifically, we have the following models to capture the interplay between human and machine decision making in the mixed autonomy.

**AV's Decision Making via $L$-step lookahead planning.** At time step $t$, after observing the current state $x(t)$, AV will first need to predict HV's future action $\hat{u}_H(t+i), i = 0, 1, 2, \cdots, L-1$ due to the unknown bounded rationality of HV. Based on this prediction, AV strives to find an action sequence that maximizes the cumulative reward with the predicted HV actions using trajectory optimization. In order to facilitate effective long-horizon reasoning, we augment the planning trajectory with a terminal value function approximation $\hat{Q}_{t-1}$, which is obtained by evaluating the policy obtained from previous time step. For convenience, we denote policy $\hat{\pi}_t$ as the solution to maximizing the

$L$-step lookahead planning objective, i.e.,

$$\hat{Q}_t(t) = \mathbf{E}[\sum_{i=0}^{L-1} \gamma^i r_A(t+i) + \gamma^L \hat{Q}_{t-1}(t+L)],$$

where $r_A(t+i) := r_A(\hat{x}(t+i), u_A(t+i), \hat{u}_H(t+i))$ and $\hat{Q}_{t-1}(t+L) := \hat{Q}_{t-1}(\hat{x}(t+L), u_A(t+L), \hat{u}_H(t+L))$. Meanwhile, $\hat{x}(t+i)$ is the state that the system will end up with if HV chose action $\hat{u}_H(t+i)$ and AV chose $u_A(t+i)$ at time step $t+i$.

Then the policy $\hat{\pi}(x(t)|u_H)$ is obtained by,

$$\hat{\pi}(x(t)|u_H) = \arg\max_{u_A(t)} \max_{\{u_A(t+l)\}_{l=1}^{L-1}} \hat{Q}_t(t), \tag{2}$$

where $u_H := \{u_H(t)\}_{t=1}^T$. It can be seen that AV's policy is conditioned on HV's policy via $\hat{u}_H(t)$.
*Remark* 3.1. Equation (2) can also be degenerated into many commonly used RL algorithms such as actor-critic and we include the discussion in Appendix E.

**HV's Decision Making with Bounded Rationality.** HV's decision making has distinct characteristics. As mentioned by the pioneering study of behavior theory [36], individuals have constraints in both their *understanding of their surroundings* and their *computational capacities*. Additionally, they face search costs when seeking sophisticated information in order to devise optimal decision rules. Therefore, we propose to model human as responding to robots actions with bounded rationality. We additionally assume HV choose the action by planning for a short time horizon, in contrast to the long horizon planning in AV's decision making. Specifically, at time step $t$, HV chooses the (sub-optimal) action by planning ahead for $N$ steps, i.e.,

$$\Phi_H(x(t), u_A(t), u_H(t)) := \sum_{i=0}^{N-1} r_H(x(t+i), u_A(t+i), u_H(t+i)) \tag{3}$$

Meanwhile, to underscore the impact of the bounded rationality in HV's decision making, we use $u_H^*(t) := \arg\max_{u_H(t)} \max_{u_H(t+1), \cdots, u_H(t+N-1)} \Phi_H$ to denote the optimal solution of Equation (3) and $u_H(t)$ to denote the sub-optimal action chosen by HV. Note that HV's policy is conditioned on AV's behavior $u_A(t)$ and we assume the time horizon $N$ is short enough such that the human can effectively extrapolate the robot's course of action, i.e., $u_A(t+i)$ is the true action taken by AV. We remark that we do not assume HV has access to the overall plan of AV but only the first few time steps. It has been shown in previous work [23] that predicting a short-term sequence of controls is manageable for human, e.g., the AV will merge into HV's lane after a short period of time.

## 4. Characterization of HV and AV's Learning Performance

### 4.1. Regret of AV with $L$-step Lookahead Planning

In this subsection, we study the impact of bounded rationality and uncertainty-aware planning on the performance of AV. To this end, we first quantify the performance gap between choosing optimal actions and sub-optimal actions, for given HV's behavior *fixed*. Therefore, conditioned on HV's action $u_H = \{u_H(t)\}_{t=1}^T$, the regret for $T$ interaction of AV is defined as

$$\mathcal{R}_A(T|u_H) = \frac{1}{T} \sum_{t=1}^T \text{Reg}_A(t) := \mathbf{E}_{x \sim \rho_0} \left[ \frac{1}{T} \sum_{t=1}^T \left( V^*(x|u_H(t)) - V^{\hat{\pi}_t}(x) \right) \right],$$

where we use $V^*(x|u_H(t))$ to denote the optimal value function attained by the optimal policy $\pi^*$ given HV's action $u_H$. $\hat{\pi}_t$ is the policy obtained in the $t$-th time step while AV solving $L$-step lookahead planning objective Equation (2) based on its prediction on HV's future actions. In particular, at each time step $t$, conditioned on HV's action $u_H(t)$, the optimal value function $V^*(x|u_H(t))$ is determined by choosing a policy $\pi_A^*(t)$ from policy space $\Pi_A$. Hence, the regret defined for AV is closely related to adaptive regret [37]. Without loss of generality, we have a general model on HV's prediction error.

**AV's Prediction of HV's Actions.** Since HV's bounded rationality is unknown to AV and the accurate model on HV is thus challenging to obtain, we assume AV's prediction of HV's action $\hat{u}_H(t+l)$ has $\epsilon_A(t)$ difference from the HV's underlying real (sub-optimal) action $u_H(t+l)$, i.e.,

$$\hat{u}_H(t+l) = u_H(t+l) + \epsilon_A(t+l), \ \ l = 0, \cdots, L, \tag{4}$$

where we assume $\epsilon_A(t) \sim \mathcal{N}(\mu_A, \sigma_A^2 I)$ to be AV's prediction error. Given the prediction on HV's actions, we first quantify the performance gap $\mathrm{Reg}_A(t)$ of AV at each time-step $t$. Then we characterize the AV's learning performance in terms of regret $\mathcal{R}_A(T|u_H)$ in the non-linear case while considering the adaptive nature of AV's learning process, e.g., the time-varying function approximation error.

**An Illustrative Example: Performance Gap in the Linear Case.** For ease of exposition, we first consider the linear system dynamics model with system parameter $A, B_H, B_A$, i.e.,

$$x(t+1) = Ax(t) + B_A u_A(t) + B_H u_H(t).$$

In the linear case, it is easy to see the resulting state transition model when AV is planning for the future steps based on the prediction of HV's action: for $l = 1, 2, \cdots$,

$$\hat{x}(t+l) = x(t+l) + \sum_{i=1}^{l} A^{i-1} B_H \epsilon_A(t+l-i), \tag{5}$$

where we denote $x(t)$ as the real state when AV choose $u_A(t)$ and HV chooses $u_H(t)$. It can be seen that due to the *error accumulation* in AV's prediction, the state transition model tends to depart from the underlying true model significantly over prediction horizon $l$.

*Remark* 4.1 (Generalization of the Prediction Error Assumption). For ease of exposition, in Equation (4), we assume the AV's prediction error follows the same distribution. While in practice, AV's prediction error may evolve over time as it accumulates more interaction history during their interactions. our analysis only requires minor modification to address the time-varying case, e.g., $\epsilon_A(t) \sim \mathcal{N}(\mu_A(t), \sigma_A^2(t)I)$. The major change lies in the joint distribution of the error accumulation term in Equation (5). The detailed steps are deferred in Appendix D.

Next, we present the performance for a single interaction, considering assumptions about function approximation error as follows.

**Assumption 4.2.** The value function approximation error in the $t$-th step is $\epsilon_{v,t}(x) := V^*(x) - \hat{V}_t(x)$ with mean $\mathbf{E}_x[\epsilon_{v,t}(x)] = \mu_{v,t}$. The value function is upper bounded by $V_{\max}$.

In practice, the optimal value function can be estimated by using Monte-Carlo Tree Search (MCTS) over a class of policies or the offline training with expert prior [38]. Denote $C_i = A^{i-1} B_H$ and $\mathcal{C}_A(l) = \|\sum_{i=1}^{l} C_i \mu_A\|_2^2 + \|\sigma_A \left(\sum_{i=1}^{l} C_i C_i^\top\right)\|_F^2$. Then, we have the following results on the performance gap in time-step $t$.

**Lemma 4.3** (AV's Performance Gap in the Linear Case.). *Suppose Assumption 4.2 holds. Then we have the following upper bound on the performance gap of AV in the $t$-th step:*

$$\mathbf{E}\left[V^*(x|u_H) - V^{\hat{\pi}_t}(x)\right] \leq \gamma^L \mu_{v,t} + \sum_{l=1}^{L} (V_{\max} + lR_{\max})\gamma^l \sqrt{\mathcal{C}_A(l)}.$$

**Error Accumulation in Planning.** In Lemma 4.3, we present a tight bound on the performance gap, where the first term in the upper bound is associated with the function approximation error and the second term is related to the AV's prediction error on HV's future action. Clearly, increasing the planning horizon $L$ can help to reduce the dependency on the accuracy of function approximation in a factor of $\gamma^L$ while risking the compounding error (the second term). Notably, the function approximation error $\mu_{v,t}$ will change during the learning process (ref. Equation (2)) and further have impact on AV's performance gap.

**Performance Gap in the Non-linear Case.** Observing the error accumulation in the linear case (ref. Equation (5)), The disparity between the actual state and the predicted state, denoted as $x(t) - \hat{x}(t)$,

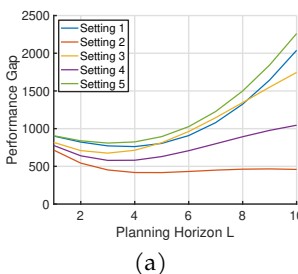
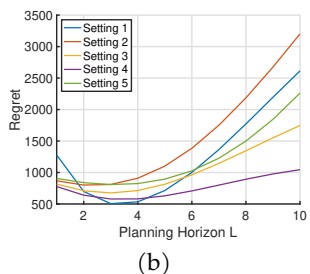
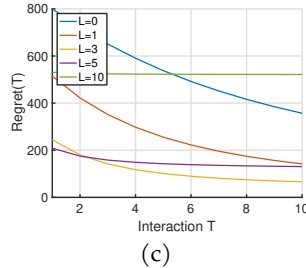

Figure 1: Numerical results on AV's regret. (a) The impact of planning horizon $L$ on AV's performance gap (ref. Lemma 4.4). (b) The impact of the planning horizon $L$ on AV's regret $\mathcal{R}_A$. (c) The impact of planning horizon on regret dynamics $\mathcal{R}_A(T)$ during the interactions.

tends to grow noticeably with time step $t$. Thus inspired, for the general case where the system model is unavailable, we formulate the prediction error as a diffusion process, i.e., denote $y(t) = x(t) - \hat{x}(t)$, then we have,

$$dy(t) = \mu_A dt + \Sigma_A^{1/2} dW(t), \; y(0) = 0,$$

where $t\mu_A$ is the drift term adn $t\Sigma_A := t\sigma_A^2 I$ is the variance term. $W(t)$ is the Weiner process.

For simplicity, let $\mathcal{E}_A(l) := \frac{(1+l)^2 l^2}{4}\|\mu_A\|_2^2 + \mathrm{tr}\left(\sigma_A^2 \frac{(1+l)l}{2}I\right)$. Then we can have the following results on the performance gap in the non-linear case.

**Lemma 4.4** (AV's Performance Gap in Non-linear Case). *Suppose Assumption 4.2 holds, then we have the upper bound of AV's performance gap in the $t$-th step as follows,*

$$\mathbf{E}\left[V^*(x|u_H) - V^{\hat{\pi}_t}(x)\right] \leq \gamma^L \mu_{v,t} + \sum_{l=1}^{L}(V_{\max} + lR_{\max})\gamma^l\sqrt{\mathcal{E}_A(l)}.$$

**Goodhart's Law and Lookahead Length.** In Lemma 4.4, we examine the performance of AV through the lens of Goodhart's law, which predicts that increasing the optimization over a proxy beyond some critical point may degrade the performance on the true objective. In our case, the planning over predicted HV actions is equivalent to the optimization on a proxy object. Increasing the planning horizon is corresponding to increase the optimization pressure. As shown in Fig. 1(a), where we plot the upper bound of the learning performance by changing different planning horizon $L$, the learning performance of AV clearly demonstrate the Goodhart's law, when increasing the planning horizon will initially help with the learning performance until a critical point. In practice, adjusting the look-ahead length (e.g., through grid search) is essential to enable AV to achieve the desired performance.

**Regret Analysis in the Non-linear Case.** To analyze the upper bound on the regret, we first impose the following standard assumptions on the MDP.

**Assumption 4.5** (Quadratic Reward Structure). The reward functions for AV and HV are the quadratic function of AV's action $u_A$ and HV's action $u_H$, respectively, i.e.,

$$r_H(x, u_A, u_H) = f_H(x, u_A) + u_H^\top S_H u_H, \quad r_A(x, u_A, u_H) = f_A(x, u_H) + u_A^\top S_A u_A,$$

where $S_H$ and $S_A$ are positive definite matrix with largest eigenvalue $s_{\max}$. $f_H$ and $f_A$ are the reward functions that capture the influence of other agent and can be non-linear.

We note that Assumption 4.5 is commonly used in the analysis of regret especially in model-based RL [39–41] and the studies in mixed traffic [10, 13]. In practice, the estimation of the parameter $S_H$ and $S_A$ can be achieved by various methods, e.g., Inverse Reinforcement Learning [10]. Based on our findings in the performance gap, we now have the following result on the regret corresponding to AV's learning performance.

Let $C = \max_{u_A} u_A \mu_A^\top (\mu_A \mu_A^\top)^{-1}$ and $M$ be the cardinality of the action space $U_A$ and $U_H$. Denote $\lambda = \sqrt{\mathrm{eig}_{\max}(C^\top S_A C)s_{\max}}$, $\Gamma := \frac{\gamma^{L+1}(1-\gamma^{T(L+1)})}{1-\gamma^{L+1}}$ and $\Lambda := \sum_{k=0}^{T}\prod_{i=0}^{k}\left(\gamma^{i(L+1)} \cdot \frac{\gamma(1-\gamma^L)}{1-\gamma}\right)$. Then we have the following result.

**Theorem 4.6** (Regret on AV's Decision Making). *Suppose Assumptions 4.2 and 4.5 hold, the regret of AV's decision making over $T$ interactions is bounded above by*

$$\mathcal{R}_A(T) \leq \sum_{l=1}^{L}(V_{\max} + lR_{\max})\gamma^l\sqrt{\mathcal{E}_A(l)} + \frac{\gamma^L}{T}\left(\Gamma\mu_{v,0} + \Lambda(s_{\max}M\sigma_A^2 + (s_{\max} + \lambda)\|\mu_A\|^2)\right),$$

**Reduce the Regret by Adjusting the Lookahead Length.** The upper bound in Theorem 4.6 is tight and it reveals the impact of the approximation error ($\mu_{v,0}$), prediction error ($\mu_A, \sigma_A$) and lookahead length $L$ on the learning performance. Specifically, we observe from the second term in the upper bound represents the *accumulation of the function approximation error*. The first term therein depends on the initial function approximation error $\mu_{v,0}$ and the last term is the compounding error due to the AV's prediction error during the $T$ times interactions. Our key observations are as follows: (1) Longer planning horizon, e.g., $L = 10$ in Fig. 1(b) and Fig. 1(c), will likely make the prediction error more pronounced and dominate the upper bound. (2) While in the case when the planning horizon is short, e.g., $L = 1$ in Fig. 1(b) and Fig. 1(c), we observe the function approximation error will likely dominate the upper bound. The empirical results provide the insights on how to adjust the lookahead length in practice. For instance, if the function approximation error is more pronounced than the prediction error, it is beneficial to use longer planning horizon $L$. The proof of AV's regret is relegated to Appendix A.

## 4.2. Regret of HV with Bounded Rationality

Given AV's action $u_A$, we define the regret for HV conditioned on AV's action $u_A$ as follows:

$$\mathcal{R}_H(T|u_A) = \mathbf{E}_{x(0)\sim\rho_0}\left[\frac{1}{T}\sum_{t=1}^{T}(\Phi_H^*(t) - \Phi(t))\right],$$

where $\Phi_H^*(t) := \Phi_H(x(t), u_H^*(t), u_A(t))$ is the optimal value and it is determined by choosing a policy $\pi_H^*(t)$ from policy space $\Pi_H$ such that $\Phi(x, \pi_A^t, \pi_H)$ is maximized. $\Phi(t) := \Phi_H(x(t), u_H(t), u_A(t))$ represents the value achieved when HV chooses sub-optimal action due to bounded rationality. For ease of exposition, we assume HV's decision making is myopic and HV's planning horizon is $N = 1$, such that $\Phi_H(x(t), u_A(t), u_H(t)) := r_H(x(t), u_A(t), u_H(t))$. Meanwhile, we assume HV makes sub-optimal decision as follows,

$$u_H(x(t), u_A(t)) = u_H^*(x(t), u_A(t)) + \epsilon_H(t)$$

where $\epsilon_H(t) \sim \mathcal{N}(\mu_H, \Sigma_H)$ is due to bounded rationality of humans and it is not known by AV.

Let $C_H = \max_{u_H} u_H\mu_H^\top(\mu_H\mu_H^\top)^{-1}$ and $\lambda_H = \sqrt{\text{eig}_{\max}(C_H^\top S_H C_H)s_{\max}}$, then we have the following results on the upper bound of HV's regret which shows the impact of bounded rationality on HV's performance. The proof of Theorem 4.7 is available in Appendix B.

**Theorem 4.7** (Regret for HV.). *Suppose Assumption 4.5 holds. Then we have the regret of HV's decision making over $T$ interactions to be bounded above by*

$$\mathcal{R}_H(T) \leq s_{\max}M \cdot \sigma_H^2 + (s_{\max} + \lambda_H)\|\mu_H\|^2$$

# 5. Regret Dynamics in Mixed Autonomy

Aiming to understand "*How do different decision making strategies impact the overall learning performance?*", especially on the impact of HV's bounded rationality on AV's performance, we study the regret dynamics in this section. More concretely, we denote the regret for the whole system as,

$$\mathcal{R}_{A-H}(T) := \frac{1}{T}\sum_{t=1}^{T}\left(\underbrace{\mathbf{E}\left[V^*(x|u_H^*(t)) - V^{\hat{\pi}_t}(x)\right]}_{(i)} + \underbrace{\mathbf{E}\left[\Phi(x(t), u_A^*(t), u_H^*(t)) - \Phi(x(t), u_A(t), u_H(t))\right]}_{(ii)}\right),$$

where $V^*(x|u_H^*(t))$ is the optimal value function when HV also takes the optimal action $u_H^*(t)$, e.g., $u_H^*(t) = \arg\max_{u_H}\Phi(x(t), u_A^*(t), u_H)$. Meanwhile $\Phi(x(t), u_A^*(t), u_H^*(t))$ is the optimal value when

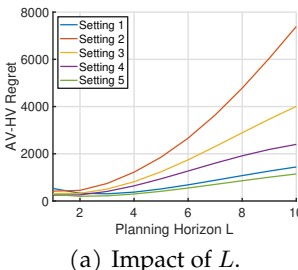
(a) Impact of $L$.

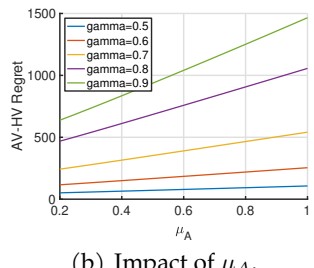
(b) Impact of $\mu_A$.

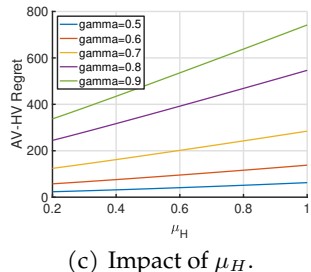
(c) Impact of $\mu_H$.

Figure 2: Empirical studies on AV and HV's decision making on the overall performance.

AV takes the optimal action $u_A^*(t) = \arg\max_{u_A} V^*(x, u_A, u_H^*(t))$ (without prediction error or function approximation error) while HV takes optimal action $u_H^*$. Intuitively, regret $\mathcal{R}_{A-H}(T)$ is defined as the difference between the best possible outcome, i.e., both AV and HV act and response to each other optimally, and the realized outcome, i.e., AV makes decision with prediction error and function approximation error while HV makes decisions with bounded rationality. Specifically, we note that the regret definition $\mathcal{R}_{A-H}$ can be naturally decomposed into two parts such that term (i) and term (ii) characterize the impact of HV's (AV's) decision making on AV (HV), respectively.

**Term (i).** Notice that term $(i)$ in $\mathcal{R}_{A-H}(T)$ can be decoupled as

$$V^*(x|u_H^*) - V^{\hat{\pi}_t}(x) := (V^*(x|u_H^*) - V^*(x|u_H)) + (V^*(x|u_H) - V^{\hat{\pi}_t}(x)).$$

The first term is induced by the sub-optimality of HV while the second term is the performance gap of AV, i.e., $\mathrm{Reg}_A(t)$.

**Term (ii).** Similarly, we can decouple term $(ii)$ into two parts, i.e.,

$$(\Phi(x(t), u_A^*(t), u_H^*) - \Phi(x(t), u_A(t), u_H^*)) + (\Phi(x(t), u_A(t), u_H^*) - \Phi(x(t), u_A(t), u_H(t))),$$

where the impact of AV's decision making is shown as the first term and the second term is the performance gap of HV, i.e., $\mathrm{Reg}_H(t)$.

Denote $\Psi_A(l) = \sqrt{\frac{(1+l)^2 l^2}{4}\|\mu_A\|_2^2 + \mathrm{tr}\left(\sigma_A^2 \frac{(1+l)l}{2} I\right)}$ and $\Psi_H(l) = \sqrt{\frac{(1+l)^2 l^2}{4}\|\mu_H\|_2^2 + \mathrm{tr}\left(\sigma_H^2 \frac{(1+l)l}{2} I\right)}$. For ease of presentation, we use notation $\Psi_v = \Gamma\mu_{v,0} + \Lambda(s_{\max} M \sigma_A^2 + (s_{\max} + \lambda)\|\mu_A\|^2)$ to represent the regret term in Theorem 4.6 and $\Xi_H = s_{\max} M \cdot \sigma_H^2 + (s_{\max} + \lambda_H)\|\mu_H\|^2$ to represent the term in Theorem 4.7. Hence, building upon our results in Theorem 4.6 and Theorem 4.7, we give the upper bound of $\mathcal{R}_{A-H}(T)$ in the following corollary.

**Corollary 5.1** (Regret of the HV-AV Interaction System). *Suppose Assumptions 4.5 holds. Then we have the upper bound on the regret of AV-HV system as follows,*

$$\mathcal{R}_{A-H}(T) \leq \sum_{l=1}^{L}(V_{\max} + lR_{\max})\gamma^l(2\Psi_A(l) + \Psi_H(l)) + \Xi_H + \frac{1}{T}\gamma^L\Psi_v$$

Corollary 5.1 shows the impact of HV and AV's decision making on the overall learning performance through terms $\Psi_A$, $\Psi_v$ and $\Psi_H$, $\Xi_H$, respectively. In what follows, we conduct the empirically studies to thoroughly examine the impact of each agent while holding another agent fixed.

**Impact of AV's decision making on the overall system performance.** (1) Implications on choosing discounting factors. Our analysis reveals three key findings regarding AV's decision-making impact. (1) First, the choice of *discount factor* significantly influences how prediction errors ($\mu_A$) affect system performance. As shown in Fig. 2(b), larger discount factors amplify the impact of prediction errors by placing greater emphasis on future rewards, evidenced by the increasing separation between performance curves at different $\mu_A$ values. (2) Second, Fig. 3(a) demonstrates that while initial function approximation error ($\mu_{v,0}$) strongly impacts regret during early interactions, its influence diminishes over time as the value function updates during learning. This aligns with the last term in Corollary 5.1. (3) Third, our results provide clear guidance on training priorities between function

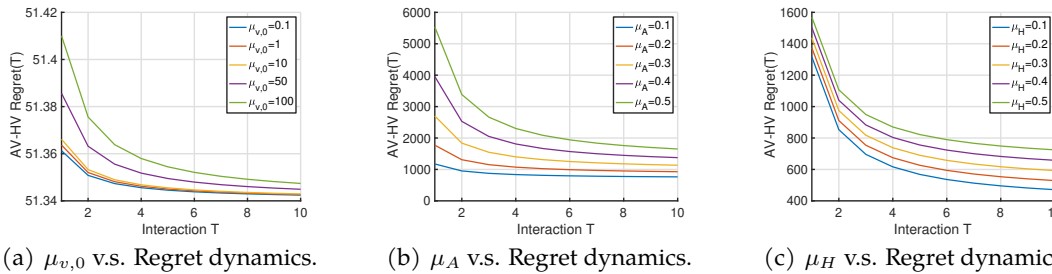

(a) $\mu_{v,0}$ v.s. Regret dynamics.  (b) $\mu_A$ v.s. Regret dynamics.  (c) $\mu_H$ v.s. Regret dynamics.

Figure 3: Empirical results on how AV and HV's decision making have impact on the overall regret dynamics, i.e., take regret as function of $T$.

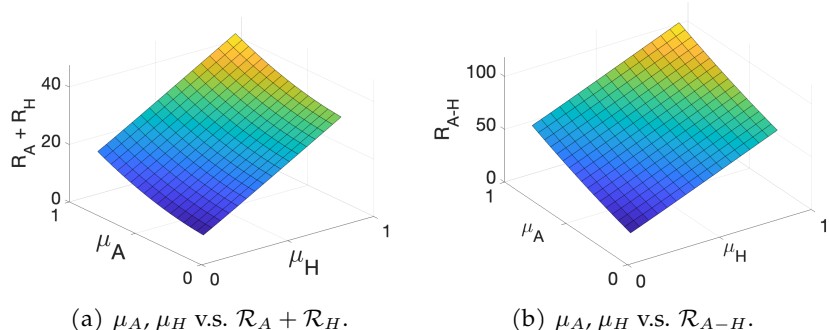

(a) $\mu_A$, $\mu_H$ v.s. $\mathcal{R}_A + \mathcal{R}_H$.  (b) $\mu_A$, $\mu_H$ v.s. $\mathcal{R}_{A-H}$.

Figure 4: Illustration of the impact of $\mu_A$, $\mu_H$ on (a) the regret summation and (b) the overall regret.

approximation and prediction model improvements. Comparing Fig. 3(a) and Fig. 3(b), reducing prediction error ($\mu_A$) from 0.4 to 0.2 yields a substantial 30% reduction in regret (from 4000 to 1800), while halving the function approximation error only reduces regret by 0.06% (from 51.41 to 51.38). This suggests that improving prediction models offers significantly greater benefits for system performance. The complete proof of these results can be found in Corollary 5.1 (Appendix C).

**Impact of HV's Bounded Rationality on the overall system performance.** As illustrated in Fig. 2(c), we conduct the experiments on the relationship between regret and human's decision making error $\mu_H$ by setting different discounting factors. In Fig. 3(c), we can see that the regret difference caused by $\mu_H$ can be consistent during the interaction, which can be related to the second term in the upper bound of $\mathcal{R}_{A-H}$. Moreover, we also demonstrate the impact of HV's decision making on AV (and vice versa) in Fig. 4. For instance, in Figure 4(b), a given $u_H$ will constrain the best possible outcome that AV can achieve, e.g., the projection on the $\mu_A$-Regret plane.

*Remark* 5.2 (Extension beyond two-agent case). Our analysis approach is feasible to extend beyond one AV and one HV setting and we outline the preliminary steps in Appendix G.

# 6. Conclusion

In this work, we take the regret analysis approach to address the questions 1) "*How does learning performance depend on HV's bounded rationality and AV's planning horizon?*" and 2) "*How do different decision making strategies between AV and HV impact the overall learning performance?*". To this end, we propose a formulation that captures heterogeneous HV-AV interactions and derive regret upper bounds for both vehicle types. Our analysis reveals two key phenomena: a Goodhart's law effect in AV's planning-based RL with predicted human actions, and error accumulation in HV's decision-making due to bounded rationality. We characterize the overall system performance through theoretical bounds and empirical studies, demonstrating the impact of different learning strategies on system efficiency.

# Acknowledgments

We acknowledge that this work is generously supported in part by NSF Grants CNS-2203239, ECCS-2413529, CCSS-2203238, and ARO Grant W911NF-2410046. We also would like to express our great appreciation to all reviewers for their constructive comments and feedback to help us improve our work.

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

# Appendix

## A. Proof of AV's Regret.

**Proxy in the System Dynamics.** In the linear case, we first derive the resulting state transition model when AV is planning for the future steps while using the prediction of HV's action. The corresponding state dynamics can be written as, i.e., after observing $x(t)$,

$$
\begin{aligned}
\hat{x}(t+1) =& A\hat{x}(t) + B_A u_A(t) + B_H \hat{u}_H(t) \\
=& Ax(t) + B_A u_A(t) + B_H u_H(t) + B_H \epsilon_A(t) \\
:=& x(t+1) + B_H \epsilon_A(t)
\end{aligned}
$$

where $x(t+1)$ is the true state when AV and HV takes action $u_A(t)$ and $u_H(t)$.

Then at the next step, we have,

$$
\begin{aligned}
\hat{x}(t+2) =& A\hat{x}(t+1) + B_A u_A(t+1) + B_H \hat{u}_H(t+1) \\
=& A\hat{x}(t+1) + B_A u_A(t+1) + B_H \hat{u}_H(t+1) \\
=& Ax(t+1) + B_A u_A(t+1) + B_H u_H(t+1) + AB_H \epsilon_A(t) + B_H \epsilon_A(t+1)
\end{aligned}
$$

It can be seen that the estimated state and the real state has the following relationship,

$$
\hat{x}(t+l) = x(t+l) + \sum_{i=1}^{l} A^{i-1} B_H \epsilon_A(t+l-i). \tag{6}
$$

**Quantify the Regret.** Recall the definition of the regret (performance gap), i.e.,

$$
\mathrm{Reg}_A(T) := \mathbf{E}_{x \sim \rho_0} \left[ \frac{1}{T} \sum_{t=1}^{T} \left( V^*(x(t)) - V^{\hat{\pi}}(x(t)) \right) \right]
$$

$$
\mathrm{Reg}_A(t) \triangleq \underbrace{V^*(x(t)) - V^{\pi_A}(x(t))}_{\text{FA Error}} + \underbrace{V^{\pi_A}(x(t)) - V^{\hat{\pi}_A}(x(t)}_{\text{Modeling Error and Lookahead}}
$$

$$
:= \underbrace{V^*(x(t)) - V^{\pi}(x(t))}_{(1)} + \underbrace{V^{\pi}(x(t)) - V^{\hat{\pi}}(x(t))}_{(2)} \tag{7}
$$

For simplicity, we define the following notations,

$\quad \hat{\tau} \quad$ trajectory obtained by running $\hat{\pi}_A$ with function approximation error (FA)

$\quad \tau \quad$ trajectory obtained by running $\pi$ with FA error

$\quad \tau^* \quad$ trajectory obtained by running in $M$ without FA error

$\quad u_t = (u_A(t), u_H(t))$

Meanwhile, we use $\hat{\pi}$ to denote the policy obtained by running lookahead on a inaccurate model and $\pi$ is the policy using the accurate model. Note that in both cases, the terminal cost are estimated by $\hat{V}$ (with function approximation error).

**Part 1. Impact of the Function Approximation Error.** We first quantify the first term (1) in Equation (7) as follows,

$$
\begin{aligned}
V^*(x_0) - V^{\pi}(x_0) =& \mathbb{E}_{\tau^*} \left[ \sum \gamma^t r(x_t, u_t) + \gamma^L V^*(s_L) \right] - \mathbb{E}_{\tau} \left[ \sum \gamma^t r(x_t, u_t) + \gamma^L V^{\pi}(x_L) \right] \\
=& \mathbb{E}_{\tau^*} \left[ \sum \gamma^t r(x_t, u_t) + \gamma^L V^*(x_L) \right] - \mathbb{E}_{\tau} \left[ \sum \gamma^t r(x_t, u_t) + \gamma^L V^*(x_L) \right] \\
& + \mathbb{E}_{\tau} \left[ \sum \gamma^t r(x_t, u_t) + \gamma^L V^*(x_L) \right] - \mathbb{E}_{\tau} \left[ \sum \gamma^t r(x_t, u_t) + \gamma^L V^{\pi}(x_L) \right] \\
=& \mathbb{E}_{\tau^*} \left[ \sum \gamma^t r(x_t, u_t) + \gamma^L V^*(x_L) \right] - \mathbb{E}_{\tau} \left[ \sum \gamma^t r(x_t, u_t) + \gamma^L V^*(x_L) \right] \\
& + \gamma^L \mathbb{E}_{\tau} \left[ V^*(x_L) - V^{\pi}(x_L) \right] \tag{8}
\end{aligned}
$$

Assumptions on the approximation error. We assume that the function approximation error is $\epsilon_v$ with mean $\mu_v$ and variance $\Sigma_v$, i.e.,

$$V^*(x) - \hat{V}(x) = \epsilon_v(x)$$

Bringing the above relation to the first two terms of Equation (8) gives us,

$$\mathbb{E}_{\tau^*}\left[\sum \gamma^t r(x_t, u_t) + \gamma^L V^*(x_L)\right] = \mathbb{E}_{\tau^*}\left[\sum \gamma^t r(x_t, u_t) + \gamma^L \hat{V}(x)(x_L) + \gamma^L \epsilon_v(x_L)\right]$$

$$\mathbb{E}_{\tau}\left[\sum \gamma^t r(x_t, u_t) + \gamma^L V^*(x_L)\right] = \mathbb{E}_{\tau}\left[\sum \gamma^t r(x_t, u_t) + \gamma^L \hat{V}(x)(x_L) + \gamma^L \epsilon_v(x_L)\right]$$

Then we have,

$$
\begin{aligned}
&V^*(x_0) - V^\pi(x_0)\\
&= \mathbb{E}_{\tau^*}\left[\sum \gamma^t r(x_t, u_t) + \gamma^L \hat{V}(x)(x_L) + \gamma^L \epsilon_v(x_L)\right]\\
&\quad - \mathbb{E}_{\tau}\left[\sum \gamma^t r(x_t, u_t) + \gamma^L \hat{V}(x)(x_L) + \gamma^L \epsilon_v(x_L)\right]\\
&\quad + \gamma^L \mathbb{E}_{\tau}\left[V^*(x_L) - V^\pi(x_L)\right]\\
&= \mathbb{E}_{\tau^*}\left[\sum \gamma^t r(x_t, u_t) + \gamma^L \hat{V}(x)(x_L)\right]\\
&\quad - \mathbb{E}_{\tau}\left[\sum \gamma^t r(x_t, u_t) + \gamma^L \hat{V}(x)(x_L)\right] + \gamma^L\left(\mathbb{E}_{\tau^*}[\epsilon_v(x_L)] - \mathbb{E}_{\tau}[\epsilon_v(x_L)]\right)\\
&\quad + \gamma^L \mathbb{E}_{\tau}\left[\hat{V}(x_L) + \epsilon_v(x_L) - V^\pi(x_L)\right]\\
&= \mathbb{E}_{\tau^*}\left[\sum \gamma^t r(x_t, u_t) + \gamma^L \hat{V}(x)(x_L)\right] - \mathbb{E}_{\tau}\left[\sum \gamma^t r(x_t, u_t)\right]\\
&\quad + \gamma^L \mathbb{E}_{\tau^*}[\epsilon_v(x_L)] - \gamma^L \mathbb{E}_{\tau}[V^\pi(x_L)]\\
&= \left(\mathbb{E}_{\tau^*}\left[\sum \gamma^t r(x_t, u_t)\right] - \mathbb{E}_{\tau}\left[\sum \gamma^t r(x_t, u_t)\right]\right)\\
&\quad + \gamma^L\left(\mathbb{E}_{\tau^*}\left[\hat{V}(x)(x_L)\right] - \mathbb{E}_{\tau}\left[\hat{V}(x)(x_L)\right]\right) + \gamma^L \mathbb{E}_{\tau^*}[\epsilon_v(x_L)]
\end{aligned}
$$

First term: (1) $\left(\mathbb{E}_{\tau^*}\left[\sum \gamma^t r(x_t, u_t)\right] - \mathbb{E}_{\tau}\left[\sum \gamma^t r(x_t, u_t)\right]\right)$.

Assume the reward function is bounded by $R_{\min} \leq r(x, u) \leq R_{\max}, \forall (x, u)$. Then we have

$$\frac{1 - \gamma^L}{1 - \gamma}(R_{\min} - R_{\max}) \leq (1) \leq \frac{1 - \gamma^L}{1 - \gamma} R_{\max}$$

Second term: (2) $\gamma^L \left(\mathbb{E}_{\tau^*}\left[\hat{V}(x)(x_L)\right] - \mathbb{E}_{\tau}\left[\hat{V}(x)(x_L)\right]\right)$. By assuming the function approximation value is bounded by $[\hat{V}_{\min}, \hat{V}_{\max}]$, we have,

$$\gamma^L(\hat{V}_{\min} - \hat{V}_{\max}) \leq (2) \leq \gamma^L \hat{V}_{\max}$$

Second term: (3) $\gamma^L \mathbb{E}_{\tau^*}[\epsilon_v(x_L)]$

$$\gamma^L \epsilon_{v,\min} \leq (3) \leq \gamma^L \epsilon_{v,\max}$$

Alternatively we have

$$(3) = \gamma^L \mu_v$$

By combing all three parts, we have the upper bound and lower bound as follows,

$$V^*(x_0) - V^\pi(x_0) \leq \frac{1 - \gamma^L}{1 - \gamma} R_{\max} + \gamma^L \hat{V}_{\max} + \gamma^L \epsilon_{v,\max}$$

$$V^*(x_0) - V^\pi(x_0) \geq \frac{1 - \gamma^L}{1 - \gamma}(R_{\min} - R_{\max}) + \gamma^L(\hat{V}_{\min} - \hat{V}_{\max}) + \gamma^L \epsilon_{v,\min}$$

**Part 2. The Impact of the Modeling Error in the $L$-step Planning.** Now we are ready to quantify the second term in Equation (7).

We first define $U_l$ as follows. For any $0 \le l \le L$, define $U_l$ to be the $l$-step value expansion that rolls out the true model $P$ for the first $l$ steps and the approximate model $\hat{P}$ for the remaining $L - l$ steps:

$$U_l = \sum_{t=0}^{l-1} \gamma^t \mathbb{E}_{x_t \sim P_t^\pi(\cdot|x)} \left[ R^\pi(x_t) \right] + \sum_{t=l}^{L-1} \gamma^t \mathbb{E}_{x_t \sim \hat{P}_{t-l}^\pi \circ P_l^\pi(\cdot|x)} \left[ R^\pi(x_t) \right]$$
$$+ \gamma^L \mathbb{E}_{x_L \sim \hat{P}_{L-l}^\pi \circ P_l^\pi(\cdot|x)} \left[ \hat{V}(x_L) \right],$$

where $\hat{P}_{L-l}^\pi \circ P_l^\pi(\cdot \mid x)$ denotes the distribution over states after rolling out $l$ steps with $P$ and $t - l$ steps with $\hat{P}$.

$$\hat{P}_{t-l}^\pi \circ P_l^\pi(\cdot \mid x) = \sum_{x' \in \mathcal{X}} P_l^\pi(x' \mid x) \hat{P}_{t-l}^\pi(\cdot \mid x')$$

Then we have,

$$U_L = V^\pi(x(t))$$
$$U_0 = V^{\hat{\pi}}(x(t))$$

Hence we have,

$$V^\pi(x(t)) - V^{\hat{\pi}}(x(t)) = U_L - U_0 = \sum_{l=0}^{L-1} U_{l+1} - U_l$$

To analyze each term in the sum, we re-arrange $U_l$ in the following ways

$$U_l = \sum_{t=0}^{l-1} \gamma^t \mathbb{E}_{x_t \sim P_t^\pi(\cdot|x)} \left[ R^\pi(x_t) \right] + \gamma^l \mathbb{E}_{x_l \sim P_l^\pi(\cdot|x)} \left[ V_{L-l}^{\hat{\pi}}(x_l) \right] \tag{9}$$

$$U_l = \sum_{t=0}^{l} \gamma^t \mathbb{E}_{x_t \sim P_t^\pi(\cdot|x)} \left[ R^\pi(x_t) \right] + \gamma^{l+1} \mathbb{E}_{x_{l+1} \sim \hat{P}^\pi \circ P_l^\pi(\cdot|x)} \left[ V_{L-l-1}^{\hat{\pi}}(x_{l+1}) \right]. \tag{10}$$

where we denote $V_L^{\hat{\pi}}(x_l) := \sum_{t=0}^{L-1} \gamma^t \mathbb{E}_{x_t \sim \hat{P}_t^\pi(x)} \left[ R^\pi(x_t) \right] + \gamma^L \mathbb{E}_{x_L \sim \hat{P}_L^\pi(x)} \left[ \hat{V}(x_H) \right]$. Note that $\hat{V}$ is not the same as $V^{\hat{\pi}}$, where the latter represents the value of running the current policy $\hat{\pi}$ with $L$ step lookahead planning over a inaccurate model with a terminal cost estimation $\hat{V}$.

Now applying Equation (10) to $U_l$ and Equation (9) to $U_{l+1}$, then we have,

$$U_{l+1} - U_l = \sum_{t=0}^{l} \gamma^t \mathbb{E}_{x_t \sim P_t^\pi(\cdot|x)} \left[ R^\pi(x_t) \right] + \gamma^{l+1} \mathbb{E}_{x_{l+1} \sim P_{l+1}^\pi(\cdot|x)} \left[ V_{\hat{P}, L-l-1}^\pi(x_{l+1}) \right]$$
$$- \sum_{t=0}^{l} \gamma^t \mathbb{E}_{x_t \sim P_t^\pi(\cdot|x)} \left[ R^\pi(x_t) \right] - \gamma^{l+1} \mathbb{E}_{x_{l+1} \sim \hat{P}^\pi \circ P_l^\pi(\cdot|x)} \left[ V_{\hat{P}, L-l-1}^\pi(x_{l+1}) \right]$$
$$= \gamma^{l+1} \mathbb{E}_{x_l \sim P_l^\pi(\cdot|x), u_l \sim \pi(\cdot|x_l)} \left[ \mathbb{E}_{x' \sim P(\cdot|x_l, u_l)} \left[ V_{L-l-1}^{\hat{\pi}}(x') \right] \right.$$
$$\left. - \mathbb{E}_{x' \sim \hat{P}(\cdot|x_l, u_l)} \left[ V_{L-l-1}^{\hat{\pi}}(x') \right] \right]$$
$$= \gamma^{l+1} \mathbb{E}_{x_l \sim P_l^\pi(\cdot|x), u_l \sim \pi(\cdot|x_l)} \left[ \int_{x'} \left( P(x' \mid x_l, u_l) - \hat{P}(x' \mid x_l, u_l) \right) V_{L-l-1}^{\hat{\pi}}(x') dx' \right]$$
$$:= \gamma^{l+1} \mathbb{E}_{x_l \sim P_l^\pi(\cdot|x), u_l \sim \pi(\cdot|x_l)} [D(x_{l+1} | P, \hat{P})],$$

where we denote $D(x_{l+1}|P, \hat{P}) = \int_{x'} \left( P\left(x' \mid x_l, u_l\right) - \hat{P}\left(x' \mid x_l, u_l\right) \right) V_{L-l-1}^{\hat{\pi}}(x')dx'$.

It can be seen that $D(x_{l+1})$ is directly relevant to the lookahead length $l$ and the modeling error $\hat{P} - P$. In the linear case, the longer lookahead length makes the difference between $P$ and $\hat{P}$ more significant, i.e., Equation (6). Next, we give the expression for $D(x_{l+1}|P, \hat{P})$ to show its relation with the lookahead length $L$.

$$D(x_{l+1}|P, \hat{P}) = \int_{x'} \left( P\left(x' \mid x_l, u_l\right) - \hat{P}\left(x' \mid x_l, u_l\right) \right) V_{L-l-1}^{\hat{\pi}}(x')dx'$$

Linear Case. Recall Equation (6),

$$\hat{x}(t + l) = x(t + l) + \sum_{i=1}^{l} A^{i-1} B_H \epsilon_A(t + l - i),$$

where $\epsilon_A \sim \mathcal{N}(\mu_A, \Sigma_A)$. Then we have,

$$\hat{P}\left(x' \mid x_l, u_l\right) = \mathbb{P}(\sum_{i=1}^{l} A^{i-1} B_H \epsilon_A(t + l - i) = x' - Ax_l - Bu_l)$$

Given $\epsilon_A$ follows a Gaussian distribution, we have

$$\sum_{i=1}^{l} A^{i-1} B_H \epsilon_A(t + l - i) \sim \mathcal{N}(\sum_{i=1}^{l} A^{i-1} B_H \mu_A, \sum_{i=1}^{l} A^{i-1} B_H \Sigma_A (A^{i-1} B_H)^\top)$$

Then we have

$$\sum_{i=1}^{l} A^{i-1} B_H \epsilon_A(t + l - i) \sim \mathcal{N}(\sum_{i=1}^{l} C_i \mu_A, \sigma_A^2 \sum_{i=1}^{l} C_i C_i^\top)$$

where $C_i := A^{i-1} B_H$.

For simplicity, assume $A^{i-1} B_H = I$, then we have

$$\sum_{i=1}^{l} A^{i-1} B_H \epsilon_A(t + l - i) \sim \mathcal{N}(l \cdot \mu_A, l\sigma_A^2 I) \tag{11}$$

Meanwhile, we have the underlying true dynamics of the system is

$$x(t + 1) = Ax(t) + B_A u_A(t) + B_H u_H(t) + \epsilon_p(t).$$

Then we have,

$$P\left(x' \mid x_l, u_l\right) = \mathbb{P}(\epsilon_p = x' - Ax_l - Bu_l)$$

Notice that $\epsilon_p \sim \mathcal{N}(0, \sigma_p^2 I)$.

Then the difference between $P$ and $\hat{P}$ boils down to the difference between two Normal distribution. We have the following results,

$$W(\hat{P}, P) = \sqrt{\|\sum_{i=1}^{L} C_i \mu_A\|_2^2 + \|(\sigma_A \left(\sum_{i=1}^{l} C_i C_i^\top\right) - \sigma_p)I\|_F^2}$$

Or in the simple case

$$W(\hat{P}, P) = \sqrt{l^2 \|\mu_A\|_2^2 + \|(\sigma_A \sqrt{l} - \sigma_p)I\|_F^2}$$

Assume the value function is bounded by $V_{\max} = \sup_h \left\| \hat{V}_l^{\hat{\pi}} \right\|_L$, i.e., the maximum Lipschitzness of the estimated value function over all possible horizons. Now we have,

$$U_{l+1} - U_l \leq V_{\max}\gamma^{l+1}\mathbb{E}_{x_{l+1}}[D(x_{l+1})] \leq V_{\max}\gamma^{l+1}\mathbb{E}_{x_{l+1}}[W(\hat{P}, P)] \tag{12}$$

where $W$ is the Wasserstein distance.

Then we have

$$U_L - U_0 \leq V_{\max}\sum_{l=1}^{L}\gamma^l\mathbb{E}_{x_{l+1},u_{l+1}\sim\pi}[W(\hat{P}(\cdot|x,u), P(\cdot|x,u))]$$

**Combining two parts gives upper bound.**

By adding the upper bound of the two parts, we obtain the upper bounds and lower bound for the performance difference,

Linear Case, no FA error. In this case, we have the regret as follows,

$$\text{Reg}_A(t) \leq \frac{1-\gamma^L}{1-\gamma}R_{\max} + V_{\max}\sum_{l=1}^{L}\sqrt{l^2\|\mu_A\|_2^2 + \|(\sigma_A\sqrt{l} - \sigma_p)I\|_F^2}$$

Linear Case, with FA error.

$$\text{Reg}_A(t) \leq \frac{1-\gamma^L}{1-\gamma}R_{\max} + V_{\max}\sum_{l=1}^{L}\sqrt{l^2\|\mu_A\|_2^2 + \|(\sigma_A\sqrt{l} - \sigma_p)I\|_F^2} + \gamma^L\epsilon_{v,\max}$$

Non-linear Case, with FA error.

$$\text{Reg}_A(t) \leq \frac{1-\gamma^L}{1-\gamma}R_{\max} + \gamma^L\hat{V}_{\max} + \gamma^L\epsilon_{v,\max}$$
$$+ V_{\max}\sum_{l=1}^{L}\gamma^l\mathbb{E}_{x_{l+1},u_{l+1}\sim\pi}[W(\hat{P}(\cdot|x,u), P(\cdot|x,u))]$$

**Treat the Prediction Error as a Diffusion Process.** Recall the diffusion process:

$$dx(t) = \mu dt + \sigma dW(t)$$
$$\text{Drift: } \mu t = \mathbb{E}[x(t) - x(0)]$$
$$\text{Variance: } \sigma^2 t = \text{Var}[x(t) - x(0)]$$

where $W(t)$ is a wiener process, i.e., $dW(t) = \varepsilon_t\sqrt{dt}$, $\varepsilon_t \sim \mathcal{N}(0, 1)$. Alternatively in the discrete case, we have $x(t) - x(0) = \mu t + \sigma W(t)$. In our setting, due to the compounding error in the lookahead planning, the difference between true state and predicted state becomes more and more different as the time horizon expands. Define the difference between the true state and predicted state as $y(t) = \hat{x}(t) - x(t)$, then we assume the prediction error follows a diffusion process, i.e.,

$$dy(t) = \mu_A dt + \Sigma_A dW(t), \ y(0) = 0$$

For simplicity, assume $\Sigma_A = \sigma_A^2 I$.

Then we can obtain that at time $t$, the prediction error follows a Gaussian distribution, i.e., $y(t) \sim \mathcal{N}(t\mu_A, t\sigma_A^2 I)$. Then we have the Wasserstein distance $\hat{P}$ and $P$ as follows [42],

$$W(\hat{P}_{l+1} - P) = \sqrt{\frac{(1+l)^2l^2}{4}\|\mu_A\|_2^2 + \text{tr}\left(\sigma_A^2\frac{(1+l)l}{2}I + \sigma_p^2I - 2\sigma_A^2\sigma_p^2\frac{(1+l)l}{2}I\right)}$$

Finally, we obtain the upper bound for the non-linear case as follows:

$$\text{Reg}_A(t) \leq \frac{1-\gamma^L}{1-\gamma} R_{\max} + \gamma^L \hat{V}_{\max} + \gamma^L \mu_{v,t}$$
$$+ V_{\max} \sum_{l=1}^{L} \gamma^l \sqrt{\frac{(1+l)^2 l^2}{4} \|\mu_A\|_2^2 + \text{tr}\left(\sigma_A^2 \frac{(1+l)l}{2} I + \sigma_p^2 I - 2\sigma_A^2 \sigma_p^2 \frac{(1+l)l}{2} I\right)}$$

**Regret over time** $T$. Now we consider the regret over time $t = 1, 2, \cdots, T$. Assume the current policy is $\hat{\pi}_t$ and the learned value function is $\hat{V}_t$. Recall that AV chose its policy in the following way,

- Estimate value function using policy $\hat{\pi}_t$:

$$\hat{Q}_{t+1} = \left(\sum_{i=1}^{L} \mathbb{E}\left[\gamma^i r_A(\hat{x}(t+i), u_A(t+i), \hat{u}_H(t+i))\right]\right.$$
$$\left. + \gamma^{L+1} \hat{Q}_t(\hat{x}(t+L+1), \hat{u}(t+L+1))\right),$$

- Derive the greedy policy (as in MPC):

$$\hat{\pi}_{t+1} = \arg \max_{u_A(t+1)} \max_{u_A(t+2), \cdots, u_A(t+L)} \hat{Q}_{t+1}$$

It can be seen that due to the update of the value function $\hat{Q}$. Next we show the difference between $\hat{Q}_{t+1}$ and $\hat{Q}_t$. Recall that we assume $V^* - \hat{V}_t = \epsilon_v$, and we denote (with abuse of notation) $Q^* - \hat{Q}_t = \epsilon_t$. Now we have

$$\hat{Q}_{t+1} - Q^* = \gamma^{L+1} \epsilon_t + \sum_{i=1}^{L} \gamma^i (\hat{r}_A - r_A),$$

where we denote $\hat{r}_A = r_A(\hat{x}(t+i), u_A(t+i), \hat{u}_H(t+i))$ and $r_A = r_A(\hat{x}(t+i), u_A(t+i), u_H(t+i))$. Similar to the analysis to HV regret, we have

$$\mu_{v,t+1} := \mathbb{E}[\epsilon_{t+1}] \leq \gamma^{L+1} \mu_{v,t} + \frac{\gamma(1-\gamma^L)}{1-\gamma}(\text{Reg}_A)$$

where $\text{Reg}_A = ms\sigma_A^2 + (s+\lambda)\|\mu_A\|^2$.

Now we are ready to derive the regret for AV as follows,

$$\text{Reg}_A(T) = \frac{1}{T} \sum_{t=1}^{T} \text{Reg}(t)$$
$$\leq \left(\frac{1-\gamma^L}{1-\gamma} R_{\max} + \gamma^L \hat{V}_{\max}\right.$$
$$+ V_{\max} \sum_{l=1}^{L} \gamma^l \sqrt{\frac{(1+l)^2 l^2}{4} \|\mu_A\|_2^2 + \text{tr}\left(\sigma_A^2 \frac{(1+l)l}{2} I + \sigma_p^2 I - 2\sigma_A^2 \sigma_p^2 \frac{(1+l)l}{2} I\right)}\bigg)$$
$$+ \frac{\gamma^L}{T}\left(\frac{\gamma^{L+1}(1-\gamma^{T(L+1)})}{1-\gamma^{L+1}} \mu_{v,0} + \sum_{k=0}^{T} \prod_{i=0}^{k}\left(\gamma^{i(L+1)} \cdot \frac{\gamma(1-\gamma^L)}{1-\gamma} \text{Reg}_A\right)\right)\bigg)$$
$$= \sum_{l=1}^{L} (V_{\max} + l R_{\max})\gamma^l \sqrt{\frac{(1+l)^2 l^2}{4} \|\mu_A\|_2^2 + \text{tr}\left(\sigma_A^2 \frac{(1+l)l}{2} I\right)}$$
$$+ \frac{\gamma^L}{T}\left(\Gamma \mu_{v,0} + \Lambda(s_{\max} M \sigma_A^2 + (s_{\max} + \lambda)\|\mu_A\|^2)\right),$$

where $\Gamma := \frac{\gamma^{L+1}(1-\gamma^{T(L+1)})}{1-\gamma^{L+1}}$ and $\Lambda := \sum_{k=0}^{T} \prod_{i=0}^{k}\left(\gamma^{i(L+1)} \cdot \frac{\gamma(1-\gamma^L)}{1-\gamma}\right)$.

## B. Proof of HV's Regret.

Due to the bounded rationality, HV does not choose the optimal action and thus introduces the regret as follows

$$\text{Reg}_H(T) := \frac{1}{T} \sum_{t=1}^{T}$$

$$\text{Reg}_H(t) = \mathbb{E}\left[\Phi(x(t), u_H^*(t), \hat{u}_A(t)) - \Phi(x(t), u_H(t), \hat{u}_A(t))\right],$$

where we assume that HV can observe the action of AV in a timely manner. Next, we impose the assumptions on the reward structure to be quadratic, i.e.,

$$r_H(x, u_A, u_H) = f_H(x, u_A) + u_H^\top S_H u_H, \tag{13}$$

where $S_H$ are positive definite matrices.

Then we have the regret for HV to be,

$$\begin{aligned}
\text{Reg}_H(t) =& \mathbb{E}\left[(u_H^*(t))^\top S_H u_H^*(t) - (u_H(t))^\top S_H u_H(t)\right] \\
=& \frac{1}{2}\mathbb{E}\Bigg[(u_H^*(t) + u_H(t))^\top S_H (u_H^*(t) - u_H(t)) \\
& + (u_H^*(t) - u_H(t))^\top S_H (u_H^*(t) + u_H(t))\Bigg] \\
=& \frac{1}{2}\mathbb{E}\left[\left((u_H^*(t) + u_H(t))^\top S_H \epsilon_H(t) + \epsilon_H(t)^\top S_H (u_H^*(t) + u_H(t))\right)\right] \\
=& \frac{1}{2}\mathbb{E}\left[\left((2u_H^*(t) + \epsilon_H(t))^\top S_H \epsilon_H(t) + \epsilon_H(t)^\top S_H (2u_H^*(t) + \epsilon_H(t))\right)\right] \\
=& \mathbb{E}\left[\epsilon_H(t)^\top S_H \epsilon_H(t) + u_H^*(t)^\top S_H \epsilon_H(t) + \epsilon_H(t)^\top S_H u_H^*(t)\right] \\
=& \text{Tr}(S_H \Sigma_H) + \mu_H^\top S_H \mu_H + u_H^*(t)^\top S_H \mu_H + \mu_H^\top S_H u_H^*(t)
\end{aligned}$$

Furthermore, we have the following assumptions on the matrices

- $\Sigma_H = \sigma_H I$, where $I$ is an identity matrix.
- The dimension of the action space is $n$
- $0 < s_{\min} \leq \text{eig}(S_H) \leq s_{\max}$, where $\text{eig}(S_H)$ is the eigenvalue of $S_H$.
- There exist a matrix $C$ such that $C_{\min}\mu_H \leq u_H^*(t) \leq C\mu_H$, notice that $u_H^*$ depends on AV's action.

With those assumptions in place, we have the upper bound for the regret as follows:

$$\text{Reg}_H(T) \leq n s_{\max} \cdot \sigma_H^2 + (s_{\max} + \lambda)\|\mu_H\|^2$$

where $\lambda := \sqrt{\text{eig}_{\max}(C^\top S_H C) \cdot s_{\max}}$.

## C. Proof of Corollary 5.1

We denote the regret for the whole system as $\mathcal{R}_{A-H}(T)$, i.e., $\mathcal{R}_{A-H}(T) :=$

$$\frac{1}{T}\sum_{t=1}^{T}\left(\underbrace{\mathbf{E}\left[V^*(x|u_H^*(t)) - V^{\hat{\pi}_t}(x)\right]}_{(i)} + \underbrace{\mathbf{E}\left[\Phi(x(t), u_A^*(t), u_H^*(t)) - \Phi(x(t), u_A(t), u_H(t))\right]}_{(ii)}\right),$$

where $V^*(x|u_H^*(t))$ is the optimal value function when HV also takes the optimal action $u_H^*(t)$, e.g., $u_H^*(t) = \arg\max_{u_H} \Phi(x(t), u_A^*(t), u_H)$. Notice that both term (i) and term (ii) can be decomposed in

the following way

$$V^*(x|u_H^*(t)) - V^{\hat{\pi}_t}(x)$$
$$= \underbrace{V^*(x|u_H^*(t)) - V^*(x|u_H(t))}_{(a)} + \underbrace{V^*(x|u_H(t)) - V^{\hat{\pi}_t}(x)}_{(b)}$$

$$\Phi(x(t), u_A^*(t), u_H^*(t)) - \Phi(x(t), u_A(t), u_H(t))$$
$$= \underbrace{\Phi(x(t), u_A^*(t), u_H^*(t)) - \Phi(x(t), u_A(t), u_H^*(t))}_{(a)} + \underbrace{\Phi(x(t), u_A(t), u_H^*(t)) - \Phi(x(t), u_A(t), u_H(t))}_{(b)}$$

In the decomposition above, term (b) is related to AV and HV's regret, respectively. Now we quantify term (a).

**AV.** Term (a) is related to $V^*$ function and we need to show that due to the bounded rationality of HV, it has direct impact on AV's overall best possible performance, i.e., denote the trajectory collected by running through MDP $M$ with HV's action $u_H^*$ as $\tau^{\mathrm{opt}}$, while the trajectory collected with HV's action $u_H$ is denoted as $\tau$, then we have

$$(a) = \mathbb{E}_{\tau^{\mathrm{opt}}}\left[\sum \gamma^t r(x_t, u_t) + \gamma^L V^*(x_L)\right] - \mathbb{E}_\tau\left[\sum \gamma^t r(x_t, u_t) + \gamma^L V^*(x_L)\right]$$
$$= \mathbb{E}_{\tau^{\mathrm{opt}}}\left[\sum \gamma^t r(x_t, u_t)\right] - \mathbb{E}_\tau\left[\sum \gamma^t r(x_t, u_t)\right] + \mathbb{E}_{\tau^{\mathrm{opt}}}\left[\gamma^L V^*(s_L)\right] - \mathbb{E}_\tau\left[\gamma^L V^*(s_L)\right]$$
$$= \sum_{i=1}^L \gamma^i \left(\eta^{i,\mathrm{opt}}(x, u) - \eta^i(x, u)\right) r(x, u) + \gamma^L \int_x \mathbb{P}[x|s_{L-1}, u_{L-1}^*] - \mathbb{P}[x|x_{L-1}, u_{L-1}]V^*(x)$$
$$\leq \sum_{i=1}^L \gamma^i \cdot i\epsilon_m r_{\max} + \gamma^L L V_{\max} \epsilon_m,$$

where $\epsilon_m$ is the total variation between $M$ and $\hat{M}$ due to HV's noisy action as the disturbance is upper bounded by $\epsilon_m$. The explicate formulation of the upper bound is available in Prop. 2.1 [43].

**HV.** Term (a) is related to $\Phi$ and we have $(a) \leq R_{\max}$

Denote $\Psi_A(l) = \sqrt{\frac{(1+l)^2 l^2}{4}\|\mu_A\|_2^2 + \mathrm{tr}\left(\sigma_A^2 \frac{(1+l)l}{2} I\right)}$ and $\Psi_H(l) = \sqrt{\frac{(1+l)^2 l^2}{4}\|\mu_H\|_2^2 + \mathrm{tr}\left(\sigma_H^2 \frac{(1+l)l}{2} I\right)}$. For ease of presentation, we use notation $\Psi_v = \Gamma\mu_{v,0} + \Lambda(s_{\max}M\sigma_A^2 + (s_{\max} + \lambda)\|\mu_A\|^2)$ to represent the regret term in Theorem 4.6 and $\Xi_H = s_{\max}M \cdot \sigma_H^2 + (s_{\max} + \lambda_H)\|\mu_H\|^2$ to represent the term in Theorem 4.7. Hence, building upon our results in Theorem 4.6 and Theorem 4.7, we give the upper bound of $\mathcal{R}_{A-H}(T)$

$$\mathcal{R}_{A-H}(T) \leq \sum_{l=1}^L (V_{\max} + lR_{\max})\gamma^l(2\Psi_A(l) + \Psi_H(l)) + \Xi_H + \frac{1}{T}\gamma^L\Psi_v$$

# D. General Setting with Time-varying Prediction Error Distribution.

**Multimodal Predictions in Autonomous Driving.** In the context of trajectory prediction in autonomous driving, multimodality arises from the fact that, given the observed information, there can be multiple plausible future trajectories for the HV. Consequently, the AV necessitates the ability to learn from the historical interactions with HV and adjust its own prediction model. Toward this end, we consider the general setting for AV's prediction error distribution, i.e., we assume the prediction error follows a time-variant distribution as follows,

$$\epsilon_A(t) \sim \mathcal{N}(\mu_A(t), \sigma_A^2(t)I), \tag{14}$$

where $\mu_A(t)$ is the time-varying mean and $\sigma_A^2(t)$ is the time-varying variance. In what follows, we demonstrate the major modification (in blue) of the proof of regret derived in the main paper.

Linear Case. Recall Equation (6),

$$\hat{x}(t+l) = x(t+l) + \sum_{i=1}^{l} A^{i-1} B_H \epsilon_A(t+l-i),$$

where $\epsilon_A(t) \sim \mathcal{N}(\mu_A(t), \Sigma_A(t))$. Then we have,

$$\hat{P}(x' \mid x_l, u_l) = \mathbb{P}(\sum_{i=1}^{l} A^{i-1} B_H \epsilon_A(t+l-i) = x' - Ax_l - Bu_l)$$

Given $\epsilon_A$ follows Gaussian distribution, we have

$$\sum_{i=1}^{l} A^{i-1} B_H \epsilon_A(t+l-i) \sim \mathcal{N}(\sum_{i=1}^{l} A^{i-1} B_H \mu_A(i), \sum_{i=1}^{l} A^{i-1} B_H \Sigma_A(i)(A^{i-1} B_H)^\top)$$

Then we have

$$\sum_{i=1}^{l} A^{i-1} B_H \epsilon_A(t+l-i) \sim \mathcal{N}(\sum_{i=1}^{l} C_i \mu_A(i), \sum_{i=1}^{l} \sigma_A^2(i) C_i C_i^\top)$$

where $C_i := A^{i-1} B_H$.

For simplicity, assume $A^{i-1} B_H = I$, then we have

$$\sum_{i=1}^{l} A^{i-1} B_H \epsilon_A(t+l-i) \sim \mathcal{N}(\sum_{i=1}^{l} \mu_A(i), \sum_{i=1}^{l} \sigma_A^2(i) I) \tag{15}$$

It can be seen that the accumulation error term Equation (15) (Equation (11)) is the major change that will affect the theoretical analysis. It is worth mentioning that in the non-linear case, we consider a special type of time-varying prediction error, i.e.,

$$\mu_A(t) = t\mu_A \tag{16}$$
$$\sigma_A^2(t) = t\sigma_A^2 \tag{17}$$

## E. Generalization of AV and HV's Learning Strategies.

**AV's Learning Strategies.** We clarify that Equation (2) can be degenerated into many commonly used RL algorithms, for instance,

- (Model-free Case) Set $L = 1$, Equation (2) is the model-free Q-function update and our regret analysis still holds.
- (Actor-Critic Case) Let $Q$-function and policy $\pi$ be parameterized by $\theta$ and $\phi$, respectively, Then Equation (2) can be learned by using Actor-Critic, i.e., in the actor step, $\theta$ is updated by maximizing the $L$-step look-ahead objective and $\phi$ is updated using policy gradient. Note that in this case, the approximation error in both Actor and Critic update can be encapsulate into $\epsilon_{v,t}$ as in Assumption 1. Our proof of the regret remains the same.

**HV's Learning Strategies.** In Equation (3), we consider AV's decision making to be $N$-step planning while we do not impose any constrains on the length of $N$. In particular, when $N \to \infty$, the decision making of HV is related to dynamic programming (assume the model is available) and otherwise, the decision making of AV is in the same spirit of Model Predictive Control (MPC).

## F. Experimental Settings.

In this section, we include the detailed parameter setup when conducting the experiments. The default setting is as follows:

- $\gamma = 0.85$

- $L = 5$

- $\mu_{v,0} = 10$

- $V_{\max} = 10$

- $R_{\max} = 1$

- $\mu_A = 1.8$

- $\sigma_A = 1$

- $M = 10$

In Figure 4 we choose the parameters as follows:

- $\gamma = 0.5$

- $T = 5$

- $V_{\max} = 10$

- $R_{\max} = 1$

- $\sigma_A = 0.1$

- $\sigma_H = 0.1$

- $s_{\max} = 2$

- $\lambda = 10$

- $M = 10$

- $l = 2$

We list the parameter settings of Figure 1(a), Figure 1(b) and Figure 2(a) in Table 1, Table 2 and Table 3, respectively.

| Parameter | Setting 1 | Setting 2 | Setting 3 | Setting 4 | Setting 5 |
|---|---|---|---|---|---|
| $\gamma$ | 0.85 | 0.85 | 0.85 | 0.85 | 0.55 |
| $\mu_{v,0}$ | 10 | 10 | 10 | 10 | 10 |
| $V_{\max}$ | 10 | 10 | 20 | 10 | 10 |
| $R_{\max}$ | 1 | 5 | 1 | 1 | 1 |
| $\mu_A$ | 0.8 | 1.8 | 1.8 | 1.8 | 1.8 |
| $\sigma_A$ | 1 | 1 | 1 | 1 | 1 |
| $M$ | 10 | 10 | 10 | 10 | 10 |

Table 1: Parameter Settings in Figure 1(a)

| Parameter | Setting 1 | Setting 2 | Setting 3 | Setting 4 | Setting 5 |
|---|---|---|---|---|---|
| $\gamma$ | 0.85 | 0.85 | 0.85 | 0.85 | 0.55 |
| $\mu_{v,0}$ | 10 | 10 | 10 | 10 | 10 |
| $V_{\max}$ | 10 | 10 | 20 | 10 | 10 |
| $R_{\max}$ | 1 | 5 | 1 | 1 | 1 |
| $\mu_A$ | 0.8 | 1.8 | 1.8 | 1.8 | 1.8 |
| $\sigma_A$ | 1 | 1 | 1 | 1 | 1 |
| $M$ | 10 | 10 | 10 | 10 | 10 |
| $T$ | 10 | 5 | 5 | 5 | 5 |

Table 2: Parameter Settings in Figure 1(b)

| Parameter | Setting 1 | Setting 2 | Setting 3 | Setting 4 | Setting 5 |
|---|---|---|---|---|---|
| $\gamma$ | 0.85 | 0.85 | 0.85 | 0.85 | 0.55 |
| $\mu_{v,0}$ | 10 | 10 | 10 | 10 | 10 |
| $V_{\max}$ | 10 | 10 | 20 | 10 | 10 |
| $R_{\max}$ | 1 | 5 | 1 | 1 | 1 |
| $\mu_A$ | 0.8 | 1.8 | 1.8 | 1.8 | 1.8 |
| $\sigma_H$ | 0.1 | 0.5 | 0.1 | 0.1 | 0.1 |
| $\sigma_A$ | 1 | 1 | 1 | 1 | 1 |
| $M$ | 10 | 10 | 10 | 10 | 10 |
| $T$ | 5 | 10 | 10 | 10 | 10 |

Table 3: Parameter Settings in Figure 2(a)

## G. Extension beyond Two Agent Case.

ur analysis approach is feasible to extend beyond one AV and one HV setting. Assume there are $N_H$ number of HVs and $N_A$ number of AVs in the mixed traffic system. With abuse of notations, we define the action vector for AVs and HVs as follows, at time step $t$, $u_H(t) = [u_{H,1}(t), u_{H,2}(t), \cdots, u_{H,N_H}(t)]$, $u_A(t) = [u_{A,1}(t), u_{A,2}(t), \cdots, u_{A,N_A}(t)]$. By defining the prediction error as in Equation (4) and HVs' bounded rationality as in Section 4.2, our analysis framework still can be applied. The dimension of the approximation error term and the bounded rationality term is thus $N_A$ and $N_H$ times higher than the two-agent case. Hence, the resulting regret in Theorem 3 and Theorem 4 are $N_A$ and $N_H$ times higher than the two-agent case.

