# OpenReview forum: "Heterogeneous Decision Making in Mixed Traffic: Uncertainty-aware Planning and Bounded Rationality"
_CPAL.cc/2025/Proceedings_Track — CPAL 2025 (Proceedings Track) Poster_

### Official Review · Reviewer_LgMH · 2025-01-09
**Theory Paper on Automated Vehicle and Human-Vehicle Interaction in a Mixed Autonomy Setting, I am not very familiar with this domain, so my opinion might not be valuable.**

**Rating:** 6
**Confidence:** 2

**Review:**

This paper study human vehicle interactions within a stochastic game framework. The authors provide regret bounds from each agent’s perspective and present empirical results that support their theoretical findings.

**Strengths**:
- The paper offers a novel theoretical perspective on mixed autonomy and vehicle-human interaction, making it a valuable contribution to the field.
- The writing is clear. Despite my limited familiarity with this domain, I was able to follow the narrative and grasp the main ideas.

**Weaknesses**:
- The paper introduces many new assumptions and settings, like "L-step lookahead planning," "Bounded Rationality," and a two-agent problem for autonomous vehicle planning. Since this is a novel analysis, it would strengthen the paper to explicitly clarify whether these assumptions are primarily for theoretical proof or are critical for broader applicability. Additionally, it would be helpful to address how easily these assumptions can be generalized.

**I am not very familiar with this domain, so my opinion might not be valuable.**

---

### Official Review · Reviewer_yAEn · 2025-01-10

**Rating:** 4
**Confidence:** 2

**Review:**

The authors study the problem of interactions, say between autonomous vehicles (AV) and human vehicles (HV), by modeling the HVs as having bounded rationality while the AVs have perfect rationality. They prove several results of the regret of the AV under a single interaction and in the long-term under a few different assumptions.

The problem setting of this paper is practically grounded, and studying the interaction between human vehicles and autonomous vehicles is rapidly becoming an important problem. However, the paper is really dense and packed with notation that's sometimes imprecisely defined (for instance: are the actions/states vectors in $\mathbb{R}^{d}$? Otherwise, how to implement the $\epsilon$ parameter in (4), etc?) Moreover, the problem statement seems to discuss several different formulations of "bounded rationality" and makes several strong simplifications (e.g. considering only one of each agent, semi quadratic objective) without arguing that these make sense in the context of the core technical problem of interactions while driving, but these arguments would form a large part of a technical contribution towards this problem of modeling driving interactions. Overall, the technical contributions are un-organized, and they seem to need strong (potentially unrealistic) simplifying assumptions to go through. It's therefore hard to understand the extent of the contributions here.

Additionally, independently of the above evaluation and the below score, I am not sure that this paper fits within the subject areas delineated by CPAL: https://cpal.cc/subject_areas/ --- but I could be wrong and so I give the review independently of this belief.

---

### Official Review · Reviewer_LVfq · 2025-01-10
**Appears rigorous but unclear significance**

**Rating:** 5
**Confidence:** 1

**Review:**

This paper uses regret analysis to study decision-making in traffic environments involving both automated vehicles (AVs) and human-driven vehicles (HVs). The authors:
- Propose a formulation for modeling AV-HV interactions,
- Derive regret upper bounds for each vehicle type,
- Use these bounds to analyze the effects of parameters such as bounded rationality and finite planning horizon in reinforcement learning (RL) settings.

Strengths:
I found the paper well-structured and easy to follow.
The theoretical analysis appears well-supported by experimental results. The experiments demonstrate how regret changes with planning horizon, degradation following Goodhart's law, the impact of bounded rationality, error dynamics, and practical implications for the planning horizon.

Weaknesses:
The main thing that I feel the work could do better is make a case for significance. Overall trends in both theoretical and experimental results seem aligned with intuition, as well as the emergence of Goodhart's law. The specifics of the bounds are dependent on assumptions like 2-agent systems and unimodal normal error in state predictions; while these are briefly discussed in the appendix the degree of realism and practical utility of the main paper results felt unclear to me.
I'm also not sure if CPAL has the right audience for this work, or how closely the paper connects to the conference's themes of structure, sparsity, parsimony, and their interaction with learning.

Overall:
Based on my comments above, I would recommend a weak reject. While the paper appears technically rigorous, it would benefit from clearer articulation of its significance, particularly with respect to the broader implications and the conference's themes.

However, I am not an expert in regret analysis or multi-agent systems and cannot comment on the novelty of the contributions or related work. Please consider my review as a non-expert perspective.


I noted some typos while reading. See below.
- line 40: "AC" should be "AV"
- line 60: "AV'" should be "AV's"
- line 90: "filed" should be "field"
- line 254: "it is beneficial of using" => "beneficial to use"
- line 463: "Bring" should be "Bringing"
- line 495: "Gaussian distribution" => "a Gaussian distribution"
- line 506: "Combing" => "Combining"

---

### Meta-Review · Program_Chairs · 2025-02-04

**Recommendation:** Reject
**Confidence:** 4

**Metareview:**

The paper investigates the interaction between automated vehicles (AVs) and human-driven vehicles (HVs) in mixed traffic environments.
It focuses on two key challenges:
a) HVs make decisions with bounded rationality, which is difficult to model accurately.
b) AVs require uncertainty-aware planning to respond safely to HV behavior.
The study aims to develop a theoretical foundation for understanding the dynamics of AV-HV interactions and improving mixed autonomy performance.

The authors analyze the learning performance of AVs and HVs by studying regret dynamics, which measure the gap between optimal and realized outcomes. The paper addresses how HVs’ bounded rationality and AVs’ planning strategies impact overall system performance. It identifies key phenomena: Goodhart’s Law: Increasing AVs’ optimization pressure (e.g., longer planning horizons) can degrade performance beyond a critical point due to prediction errors; Error Accumulation: HVs’ bounded rationality leads to compounding effects in decision-making, impacting both AV and HV performance

### Pros
* This paper provides the first theoretical framework for analyzing heterogeneous decision-making in mixed autonomy traffic, with both autonomous vehicles and human drivers.
* It provides meaningful insights for mixed systems like Goodhart’s Law and Error Accumulation, based on assumptions or insights of behavioral economics like bounded rationality.

### Cons
* The audience within CPAL for this paper may be limited, as noted by some reviewers.
* The technical contributions are not sufficiently highlighted in the paper.
* The reviewers raised several points on the quality of the current manuscript.

The mixed opinions among reviewers and their relatively low confidence levels make it hard to recommend this conference for acceptance at CPAL.

---

### Decision · Program_Chairs · 2025-02-11

Accept (Poster)